# Influence of Body Dissatisfaction on the Self-Esteem of Brazilian Adolescents: A Cross-Sectional Study

**DOI:** 10.3390/ijerph17103536

**Published:** 2020-05-18

**Authors:** Francisco Nataniel Macêdo Uchôa, Natalia Macêdo Uchôa, Thiago Medeiros da Costa Daniele, Romário Pinheiro Lustosa, Paulo Roberto de Castro Nogueira, Victor Machado Reis, Joaquim Huaina Cintra Andrade, Naira Figueiredo Deana, Ágata Marques Aranha, Nilton Alves

**Affiliations:** 1Universidade de Trás-os-Montes e Alto Douro, 5001-801 Vila Real, Portugal; nataniel4@hotmail.com (F.N.M.U.); aaranha@utad.pt (A.M.A.); 2Centro Universitário da Grande Fortaleza (UNIGRANDE), Fortaleza 60525-571, Brazil; nataliamacedouchoa@hotmail.com (N.M.U.); paulonogueira@unigrande.edu.br (P.R.d.C.N.); 3Universidade de Fortaleza (Unifor), Fortaleza 60811-905, Brazil; danielethiago@yahoo.com.br; 4Universidade Estadual do Ceará, Fortaleza 60714-903, Cerá, Brazil; romario-lustosa@hotmail.com (R.P.L.); joaquimcintra@hotmail.com (J.H.C.A.); 5Research Center in Sports Sciences, Health Sciences & Human Development (CIDESD), 5001-801 Vila Real, Portugal; victormachadoreis@gmail.com; 6Center for Research in Epidemiology, Economics and Oral Public Health (CIEESPO), Faculty of Dentistry, Universidad de La Frontera, Temuco 4780000, Chile; n.figueiredo01@ufromail.cl; 7Applied Morphology Research Center (CIMA), Faculty of Dentistry, Universidad de La Frontera, Temuco 4780000, Chile; 8Center of Excellence in Surgical and Morphological Research (CEMyQ), Faculty of Medicine, Universidad de La Frontera, Temuco 4780000, Chile

**Keywords:** body dissatisfaction, self-esteem, adolescents

## Abstract

Background: The present study investigated the influence of body dissatisfaction (BD) on the self-esteem of Brazilian adolescents. Methods: A cross-sectional study was carried out with 1011 students at public and private schools in the city of Fortaleza, Brazil. The body shape questionnaire and the Rosenberg self-esteem scale were applied. Chi-square test, Student’s t-test, Pearson’s correlation, the odds ratio and binary logistic regression were used. Results: The rate of low self-esteem was 33.8% in the adolescents; 27.8% of the adolescents presented some degree of BD, with severe BD in 5.8%. A significant low negative correlation was found between self-esteem and BD in all the adolescents. In the Odds Ratio analysis, it was observed that the odds of having low self-esteem increased in adolescents with BD as compared to adolescents without BD, being 3.85 times higher in females (CI 95%, 2.12–6.99), 2.83 times higher in males (CI 95%, 1.22–6.58), 5.79 times higher in adolescents attending public schools (CI 95% 2.06–16.26), and 2.96 times higher in adolescents attending private schools (CI 95%, 1.79–4.88). Conclusions: Low self-esteem affected one-third of the adolescents, both male and female. BD and education in public schools are predictor variables of low self-esteem in adolescents.

## 1. Introduction

Self-esteem is considered one of the principal predictors of good results in adolescence, with implications for areas like interpersonal relations and academic performance [1]. Self-esteem can be defined as a positive or negative attitude to oneself [2]. Positive self-esteem reduces the susceptibility of adolescents to body dissatisfaction (BD) [3]. Low self-esteem is related to feelings of uselessness and failure, which can make adolescents more susceptible to dissatisfaction with their weight, physical appearance and body shape [4]. There are various specific sources of self-esteem, one of which is appearance [5]. Adolescence is a period of changes, development and establishing an identity, with a constant focus on external physical appearance [6]. In adolescence, a variety of social and corporal changes occur during and after puberty, which can have a strong impact on the individual’s body image and become more pronounced at the end of adolescence [7]. Body image refers to a multidimensional construct which includes perception, emotion, feelings and thoughts directed towards one’s own body [8]. BD is also related to dissatisfaction with one’s weight, appearance and body shape. Its prevalence among adolescents is 30–40% [4]; this percentage can vary up or down depending on sex or the population studied. Ganesan et al. [9] carried out a study with 1200 female students in the region of Coimbatore, India, in which they observed that 77.6% of the students presented some level of BD. Fortes et al. [4] carried out a study in female Brazilian adolescents from the south-east of the country, reporting that 30% presented BD and that 16% of the body dissatisfaction of the adolescents could be explained by feelings of satisfaction and self-esteem. Some studies suggest that female adolescents are much more affected by concern over their physical appearance, which may explain their reduced levels of self-esteem in this stage of life [10,11,12,13]. BD has been reported as an important predictive factor for low self-esteem and depressive mood [6,10], which are quite frequent among adolescents in western society. However, there is as yet no consensus in the literature, since other studies have failed to establish a relationship between self-esteem and BD [14]. Due to the potential negative impact of low self-esteem on the health of adolescents, it is important to learn more regarding the factors that may lead them to develop this condition, and to discover which are most susceptible in order to intervene early by establishing preventive and/or treatment programmes for this population. In response to this need, in the present study we investigate the relationship between body satisfaction and self-esteem in Brazilian adolescents of the city of Fortaleza, Ceará, Brazil.

## 2. Method

An observational cross-sectional study was carried out. The present study forms part of another investigation carried out by the same team which investigated the influence of the mass media and body dissatisfaction on the risk of developing food disorders in adolescents [11], and the impact of physical activity on body mass index and self-esteem [15].

All the students of the participating schools were invited to take part. First, a meeting was held to tell the participants the purpose of the study and their individual participation. They were told that participation in the study was voluntary and that if they did not wish to participate it would not affect their relations with their teachers or the school. Both the adolescents and their parents or guardians were asked to provide written informed consent. Once it had been accepted and signed (by the parents or guardians), the adolescents were enrolled in the study. All procedures performed in studies involving human participants were in accordance with the ethical standards of the institutional and/or national research committee and with the 1975 Helsinki Declaration and its later amendments or comparable ethical standards. This study was approved by the Ethics Committee for Research on Human Beings, file number 2.193.376.

### 2.1. Participants

This was a descriptive, observational, cross-sectional study. The sample size was calculated using the formula $n_{0}=\frac{1}{E_{0}^{2}}$, with a sample error of 4% (*n* = 621 participants). The study included 1011 secondary school students in the city of Fortaleza, Brazil, aged between 14–18 years. Adolescents from private schools (306 girls and 224 boys) and public schools (221 girls and 260 boys) were included.

### 2.2. Evaluation of Satisfaction with Body Image

To measure satisfaction with body image (BI), the body shape questionnaire (BSQ) [16] was used in the version validated in Portuguese [17]. The questionnaire contains 34 questions on a 6-point Likert scale, ranging from 1 (never) to 6 (always), spread over four sub-scales: (1) Self-perception of body shape (22 questions); (2) Comparative concern (5 questions); (3) Attitude (5 questions), and (4) Severe alterations (2 questions), to give a score ranging between 34 to 204 points. The lower the score, the greater the body satisfaction. The adolescents were divided into four levels of dissatisfaction with physical appearance [17]: no dissatisfaction with BI (score ≤ 79); slight dissatisfaction with BI (score ≥ 80 ≤ 109); moderate dissatisfaction with BI (score ≥ 110 ≤ 140); and severe dissatisfaction with BI (score ≥ 140).

### 2.3. Evaluation of Self-Esteem

To evaluate self-esteem, the Rosenberg self-esteem scale (RSS) [2] was used, validated for the Brazilian population by Hutz and Zanon [18]. This scale consists of ten statements related to a set of feelings of self-esteem and self-acceptance, which indicate overall self-esteem. The responses to the items are given on a four-point Likert-type scale: totally agree, agree, disagree and totally disagree. The scores were categorised into: ≤25 points, low self-esteem; 26 to 29 points, medium self-esteem; and 30 to 40 points, high self-esteem.

### 2.4. Procedures

To complete the questionnaires, the adolescents sat in a classroom in the presence of one of the study investigators previously trained to support students and address any questions or doubts which they may have had. The participants took between 20 and 30 min to complete all the questions. Then another investigator was responsible for measuring their weight and height.

### 2.5. Data Analysis

Levene’s test of homogeneity of variances and the Kolmogorov–Smirnov test were carried out to analyse the normality of the data. Descriptive statistics are presented as mean ± standard deviation (±SD). The chi-square test was used for qualitative variables. Student’s t-test was used for comparison between the sexes and between schools. ANOVA was applied to compare the levels of BD and self-esteem. Pearson’s correlation and the Odds Ratio were applied between the variables analysed. A binary logistic regression was carried out to assess whether body image, sex, age and school type could predict the level of self-esteem of the adolescents. The Figures were made with the GraphPad Prism programme, version 8.0 (GraphPad Software Inc, San Diego, CA, USA). Statistical analysis used SPSS for Mac, version 22.0 (IBM Chicago IL, USA). Statistical significance was set at *p* < 0.05.

## 3. Results

The study included 527 (52.1%) female adolescents and 484 (47.9%) males (*p* = 0.056); 530 (52.4%) attended private schools and 481 (47.6%) public schools (*p* = 0.029). The mean age of the participants was 15.6 years for the females and 15.7 years for the males (*p* = 0.105). The mean general RSS score was 27.9 (SD = 5.0); 26.7 (SD = 4.7) for private school students and 29.2 (SD = 5.1) for public school students (*p* < 0.001). The mean general BSQ score was 71.1 (SD = 32.4), 69.4 (SD = 31.6) for private school students and 73.0 (SD = 33.3) for public school students (*p* < 0.076).

In adolescents from private schools, it was observed that the mean score in the RSS was similar between the sexes, with no statistically significant differences, as outlined in Table 1. It was also observed that the mean score in the BSQ was significantly higher among female adolescents than males (Table 1). Among female adolescents, it was observed that private school students presented higher scores in the BSQ, showing greater body dissatisfaction than girls from public schools (*p* ≤ 0.001). It was also observed that the private school pupils’ scores in the RSS questionnaire were significantly higher, showing that they have higher self-esteem than adolescents from public schools (*p* ≤ 0.001). The same was found among male adolescents, with significantly higher RSS scores among private school pupils than public school students (*p* ≤ 0.001). It is important to note that only girls in public schools were classified with low self-esteem; all the other groups were classified with moderate self-esteem. Furthermore, only girls from public schools presented mean values compatible with the classification of slight dissatisfaction with BI, the other groups presented mean scores lower than 89, being classified as having no dissatisfaction with BI.

It was observed that 64.3% of the adolescents, of both sexes, presented moderate or low self-esteem; 34.3% were girls, and 30% were boys. In separate analyses by sex, it was observed that 34.3% of the female adolescents presented low self-esteem, 29.3% moderate self-esteem and 36.4% high self-esteem. A higher percentage of adolescents with low self-esteem was observed among the boys than among the girls, with 37.4%; moderate self-esteem was observed in 31.6% and high self-esteem in 31%. However, no statistical differences were found between the sexes (*p* = 0.187).

Twenty-eight point seven percent (28.7%) of the adolescents presented some degree of BD. Dissatisfaction with body image was significantly higher among girls (37.8%) than among boys (18.8%; *p* ≤ 0.001). It was observed that 19% of the female adolescents presented slight BD, 10.2% moderate BD and 8.7% severe BD. Of the male adolescents, 11% presented slight BD, 5.4% moderate BD and 2.5% severe BD. Severe BD affected 5.8% of the total sample of adolescents, being most frequent in female adolescents from public schools.

### 3.1. Analysis of Level of Self-Esteem Among Adolescents

Low self-esteem was observed to be more frequent among public school students than private school students (girls *p* ≤ 0.001; boys *p* ≤ 0.001). A difference between sexes was observed only for public school adolescents, where high self-esteem among girls was higher than among boys (*p =* 0.029) (Figure 1).

The highest proportion of high self-esteem was found among adolescents from private schools, of both sexes. The lowest proportion of high self-esteem was observed among adolescents from public schools, especially among females. Low self-esteem was higher in public school students of both sexes (Figure 1 and Figure 2). Among male adolescents, the mean for private school boys was significantly higher than the mean found for public school boys (Table 1).

### 3.2. Analysis of Degree of Body Dissatisfaction Among Adolescents

Absence of body dissatisfaction was higher among female than male public school students. Severe dissatisfaction, expressed by the BSQ score, was highest among females from public schools and males from private schools. When adolescents from different school types were compared, males from private schools were observed to present a higher BSQ score than males from public schools (Figure 3).

A high percentage of adolescents of both school types and both sexes were observed to be free of BD; the percentage was highest among male adolescents from private schools (Figure 4). A higher percentage of male adolescents than female did not present BD, both in public schools (*p* ≤ 0.001) and private schools (*p* ≤ 0.001), therefore BD was more frequent in girls than boys. Adolescents from private schools presented a higher percentage of body satisfaction than those from public schools (girls, *p* ≤ 0.001; boys, *p* = 0.005). It was also observed that severe BD was most frequent in female adolescents from public schools (Figure 4).

### 3.3. Analysis Between Level of Self-esteem and Degree of Body Dissatisfaction

In both sexes, low self-esteem was observed to be greater in adolescents from public schools than those from private schools, except for adolescents with severe BD, as shown in Table 2. Low self-esteem was also found to be more frequent in adolescents of both sexes from private schools, even though these adolescents presented no BD or only slight BD. The same was not true of adolescents from public schools, who presented a high percentage of high self-esteem even when their body dissatisfaction was moderate or severe (Table 2).

In the Pearson’s correlation analysis, a significant low negative correlation was found between self-esteem and BD for all adolescents: private school girls (*r =* −0.273, *p* = 0.002), public school girls (*r =* −0.234, *p* < 0.001), private school boys (*r =* −0.209, *p* < 0.001) and public school boys (*r =* −0.221, *p* < 0.001).

It was observed that female adolescents with BD had 3.95 (Confidence Interval, CI 95%, 2.12–6.99) times higher odds of presenting low self-esteem than females without BD. In male adolescents with BD, the odds of presenting low self-esteem were 2.83 (CI 95%, 1.22–6.58) times higher than in male adolescents without BD. In adolescents from public schools with BD, the odds of having low self-esteem were 5.79 (CI 95%, 2.06–16.26) times higher than in students without BD; while in private schools, adolescents with BD had 2.96 CI 95%, 1.79–4.88) times higher odds of presenting low self-esteem than adolescents without BD.

The logistic regression model showed a good fit with the statistics of Hosmer–Lemeshow (*p =* 0.446). Only level of body satisfaction and private/public school were predictor variables of self-esteem in adolescents (Table 3). The model presented a percentage correction of 67.4%, which was significant, with [X^2^(7) = 6.842; *p* < 0.001; Negelkerke R^2^ = 0.108]; this showed that BD and coming from a public school are significant predictor variables of low self-esteem in adolescents.

## 4. Discussion

The present study compared the levels of self-esteem and body satisfaction in adolescents from public and private schools in the city of Fortaleza, north-eastern Brazil. The results showed an association between low self-esteem and BD, corroborating previous studies which have reported that high levels of perceived body dissatisfaction are related with lower levels of self-esteem [19]. BD was a significant predictor variable for low self-esteem in adolescents in the present study, as adolescents with BD presented 2.31 times higher odds of presenting low self-esteem than adolescents without BD. The results of the present study corroborate those of Fortes et al. [4], showing that adolescents with low self-esteem have greater body dissatisfaction than those with high self-esteem. The association between these variables (BD and low self-esteem) was established by a longitudinal study which followed adolescents of both sexes over 5 years [10]. In their study, Paxton et al. [10] concluded that BD is a risk factor for increased depressive mood and low self-esteem in adolescents of both sexes. In a longitudinal study, Tiggemann et al. [20] found correlations between BD and low levels of self-esteem, and between greater body satisfaction and higher self-esteem, in female adolescents, however, these authors report that there is no evidence that self-esteem is a cause of the development of BD. In another study, the authors observed a significant negative correlation between dissatisfaction with body weight and self-esteem, but only in females, not in males [21]. Harter [6] found a moderate correlation between body satisfaction and self-esteem in North Americans (0.65) and individuals of other countries (0.62). In the present investigation, lower self-esteem was correlated with greater BD, with a significant low negative correlation.

Previous studies have reported various factors associated with low self-esteem in adolescents, including female sex [22], low socioeconomic level [22], overweight [23] and the negative influence of the mass media [11,12]. Some authors report that the sex of the individual is a predictor of lower self-esteem, with men presenting higher self-esteem than women [19,22]. In the present study, among public school students, females presented a lower RSS score than males, suggesting that self-esteem was lower among girls than boys. However, in the binary logistic regression model sex was not a significant predictor variable, possibly because the male adolescents included in the present study presented a high percentage with low self-esteem (37.4%); this was similar to the value among female adolescents (34.3%), showing that low self-esteem does not have a greater impact on females than males, but affects both sexes. This finding is very important for the implementation of health policies related to self-esteem among adolescents. Previously, much more attention was paid to girls than boys; however, these results show that boys are equally affected by this condition, and prevention and treatment programmes need to focus on both sexes.

Previous studies have reported a high degree of BD among adolescents [12]. A prevalence of approximately 30% in female adolescents has been reported in south-eastern Brazil [4]; this is similar to the percentage found in the present study, carried out in adolescents of both sexes in north-eastern Brazil, in which 28.7% of adolescents were found to present some degree of BD. Body dissatisfaction was more frequent among girls; it affected more than one-third of the female population studied (38.8%) while only 18.8% of boys presented some degree of BD. Moderate to severe BD affected 18.8% of female adolescents and 8.96% of male adolescents; this condition was more frequent in adolescents from public schools than those from private schools. Studies indicate that girls are more affected by the mass media due to the imposition of an intangible pattern of beauty characterised by a very slender body. This has a negative effect on female adolescents, who perceive a wide difference between their physical appearance and the idealised appearance, leading to dissatisfaction with their own bodies [11,12]. Males are also affected by imposed standards of beauty, but in a different way to females. Whereas girls aspire to more slender bodies, boys seek to develop more muscular bodies with well-defined muscles; however, they appear to be less affected by the differences between their own bodies and the “ideal body” [21]. In the present study, the odds ratio analysis showed that female adolescents with BD had a higher chance of presenting low self-esteem (OR = 3.85) than females without BD. The same was observed for male adolescents with BD who presented 2.83 times higher odds of having low self-esteem than males without BD. When the students’ school of origin was analysed (public or private), it was found that the odds of an adolescent with BD having low self-esteem was higher among public school students than private school students. Among adolescents from public schools with BD, the odds of presenting low self-esteem were 5.79 times higher than in adolescents without BD; among adolescents from private schools with BD, the odds of presenting low self-esteem were 2.96 times higher than in adolescents without BD.

In a previous study, Raymore et al. [22] analysed the relation between self-esteem in adolescents and their socioeconomic level through the association between parental education and family income; they found a significantly higher level of self-esteem among the children of parents with a higher level of education, establishing that socioeconomic status was positively related with self-esteem. The results of the present study showed that low self-esteem was much more frequent in adolescents from public schools than those from private schools. The binary logistic regression model showed that the variable ‘adolescents from public schools’ was a predictor for low self-esteem since adolescents from public schools have 2.66 times higher odds of presenting BD than students from private schools. This association between school type and self-esteem may be related with the fact that in Brazil, students at public schools come from families with less acquisitive power, lower resources, less recreation time and less educated parents/guardians, indicating that public school students have a lower socioeconomic status. Other studies have suggested that a lower socioeconomic status is capable of causing mental and physical health problems [24]; it is, therefore, an important variable when analysing the self-esteem of adolescents.

Adolescence is a complex transition stage, in which individuals may be affected by different conditions capable of a negative influence on their emotional state, determining lower self-esteem. In the present study, associations were found between BD and low self-esteem and between adolescent students from public schools and low self-esteem. A percentage of 10.8% can explain low self-esteem due to BD and coming from a public school. Future studies could explore further the relationship between socioeconomic status and self-esteem in Brazilian adolescents, addressing not only the financial aspect but also the parents’ education, interpersonal relations, acceptance and the forms of recreation to which the different economic classes have access. It must be stressed that the high percentage of low self-esteem found in the present study (33.8%) is a warning of a situation which affects many adolescents, and the other risk factors also responsible for determining low self-esteem should be investigated in greater depth. Identifying the risk factors is fundamental since once they are known, early intervention measures can be established to prevent or control this psychological alteration, as well as promoting healthier development during adolescence.

### Study Limitations

One limitation of the present study is that the adolescents were not evaluated by a specialist to confirm the finding of low self-esteem. Another limitation is in the design of the study since measurements were only taken at the time the study was applied; it was therefore not possible to establish a relation of cause and effect. It should also be noted that the sample included in the present study is only representative of the city of Fortaleza, but not of Brazil. Future studies should be carried out in large population centres in other regions of Brazil, since the self-esteem of adolescents from other regions may be more or less affected by BD or other variables inherent in each part of the country.

## 5. Conclusions

Body dissatisfaction was more frequent among girls, 38.8% of whom were affected, while only 18.8% of boys presented this condition. There is a significant low negative association between self-esteem and body dissatisfaction. Adolescents with body dissatisfaction have a higher chance of presenting low self-esteem than adolescents without body dissatisfaction; moreover, in Brazil, the chance of adolescents presenting low self-esteem is higher when they are students at a public school than at a private school. Body dissatisfaction and the fact that the adolescent is a student at a public school were predictor variables of low self-esteem in adolescents. Low self-esteem affected one-third of adolescents (33.8%), both male and female, showing that the implementation of health policies for prevention and/or treatment of self-esteem problems in adolescents should not focus only on females, but today they should be applied to both sexes.

## Figures and Tables

**Figure 1 ijerph-17-03536-f001:**
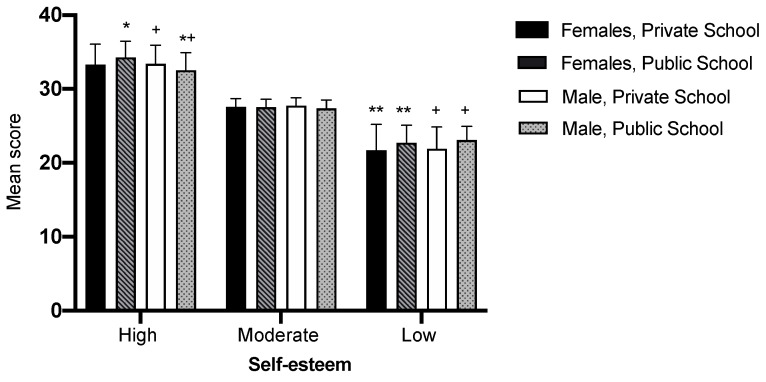
Level of self-esteem, mean and standard deviation, by sex and school type. *—*p < 0.05* statistically significant difference between sexes; +—*p < 0.05* statistically significant difference between schools, for males; **—*p < 0.05* statistically significant difference between schools, for females.

**Figure 2 ijerph-17-03536-f002:**
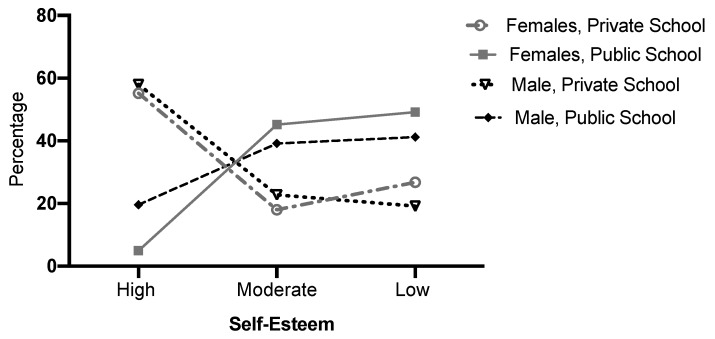
Level of self-esteem by sex and school type, expressed in percentages.

**Figure 3 ijerph-17-03536-f003:**
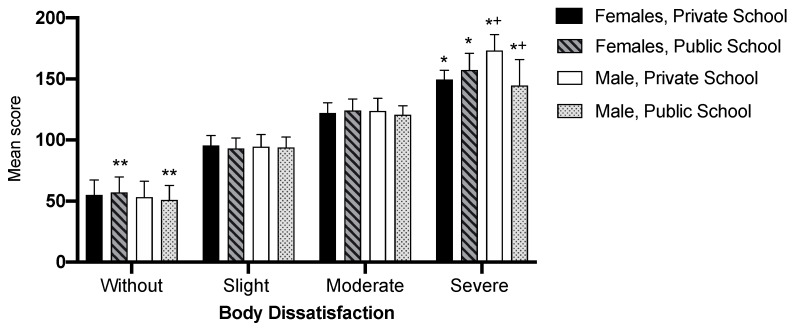
Degree of body dissatisfaction mean and standard deviation, by sex and school type. *—*p < 0.05* statistically significant difference between sexes; **—*p ≤ 0.001* statistically significant difference between sexes; +—*p < 0.05* statistically significant difference between schools.

**Figure 4 ijerph-17-03536-f004:**
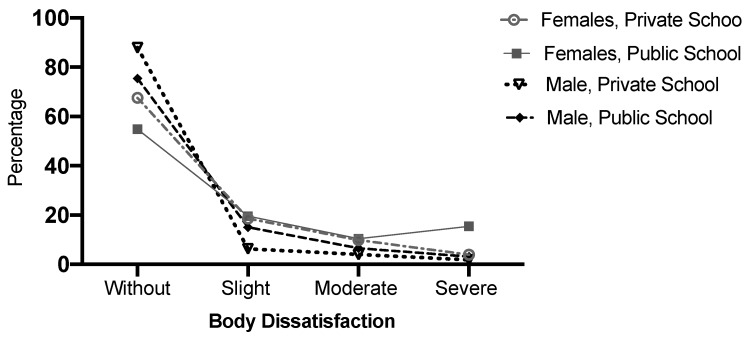
Degree of body dissatisfaction by sex and school type, expressed in percentages.

**Table 1 ijerph-17-03536-t001:** Mean values reported for the variables analysed.

School Type	Variable	Female	Male	*p*-Value
Private school	RSS score	29.15	29.91	0.114
	BSQ score	72.89	60.78	<0.001 *
Public school	RSS score	25.48	26.60	0.001 *
	BSQ score	96.51	64.84	<0.001 *

RSS—Rosenberg self-esteem scale; BSQ—Body Shape Questionnaire; *—statistically significant difference between sexes.

**Table 2 ijerph-17-03536-t002:** Level of self-esteem by degree of body satisfaction, sex and school type, expressed in percentages.

Sex	Degree of BD	Public School			Private School		
		HSE	MSE	LSE	HSE	MSE	LSE
Female	No BD	48.8%	43.8%	7.4%	20.3%	19.3%	60.4%
	Slight	41.9%	53.5%	4.7%	33.3%	14.0%	52.6%
	Moderate	52.2%	47.8%	0.0%	40.0%	23.3%	36.7%
	Severe	61.8%	38.2%	0.0%	75.0%	0.0%	25.0%
Male	No BD	37.8%	39.3%	23.0%	15.2%	24.4%	60.4%
	Slight	48.7%	41.0%	10.3%	50.0%	7.1%	42.9%
	Moderate	58.8%	29.4%	11.8%	33.3%	11.1%	55.6%
	Severe	50.0%	50.0%	0.0%	75%	25.0%	0.0%

BD—body dissatisfaction; HSE—high self-esteem; MSE—moderate self-esteem; LSE—low self-esteem.

**Table 3 ijerph-17-03536-t003:** Binary logistic regression model, with the response to low self-esteem in adolescents.

	B	SE	Wald	df	*p*-Value	OR	95% CI for OR	
							Lower	Higher
Age band	–0.268	0.139	3.690	1	0.055	0.765	0.582	1.005
School	0.981	0.141	48.072	1	<0.001	2.667	2.021	3.519
BSQ score	0.841	0.195	18.565	1	<0.001	2.318	1.581	3.398
Sex	0.267	0.143	3.503	1	0.061	1.307	0.987	1.729
Constant	–1.295	0.156	68.840	1	<0.001	0.274		
Model fit *	0.446							

B—regression coefficient; df—degrees of freedom; SE—standard error; CI—Confidence Interval OR—Odds Ratio; *—Hosmer–Lemeshow statistics.

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
