# Peer review of "Influence of Body Dissatisfaction on the Self-Esteem of Brazilian Adolescents: A Cross-Sectional Study"

_ijerph, 2020, doi:10.3390/ijerph17103536_

Round 1

Reviewer 1 Report

Introduction:

  • Replace ‘relation’ with ‘relationship’

Methods:

  • Should be third person rather than first person language – this is observed elsewhere and should be avoided. Please use third person.
  • Methods introduction should also describe the type of study i.e. longitudinal, cross sectional etc. Please provide more detail – I can see some of this in under ‘participants’, please move this under the methods introduction\
  • ‘Both the adolescents and their parents or guardians were given the informed consent’ is better described as ‘Both the adolescents and their parents or guardians were asked to provide written informed consent’
  • ‘trained to clear up all their doubts’ should read ‘trained to support students and address any questions or doubts which the students may have had’

Results

  • ‘Table 1 shows the RSS and BSQ scores, by sex and school type. In adolescents from private schools, we observed that the mean score in the RSS was similar between the sexes, with no statistically significant differences.’ Should read ‘In adolescents from private schools, we observed that the mean score in the RSS was similar between the sexes, with no statistically significant differences, as outlined in Table 1’. Best to not start a paragraph with Table xxx… This is also observed for Figure 1, Figure 3 – please change this
  • Figure 2 – the x-axis labels are cumbersome – change to High, Medium and Low only with 1 overarching label ‘Self-esteem’, much like the y-axis label ‘percentages’ (Do the same for all figures)
  • Change ‘We further observed that the chance of a female adolescent with BD presenting low self-esteem was 3.85 times higher (CI 95% 2.12-6.99) and 2.83 times higher for males (CI 95% 1.22-6.58)’ to be ‘It was observed female adolescents with BD had 3.85 (CI 95% 2.12-6.99) times higher odds of presenting with low self-esteem than females without BD, which was observed to be 2.83 (CI 95% 1.22-6.58) times higher among males with BD. – This is an issue as it is not clear, but what I think you are saying is if we have two females, one with BD one without, then the Female with BD is 3.85 times more likely to and Esteem issues that the other female without BD?
  • Please ensure the above statement is correct in the text of the paper and ensure all other statements made like the above are corrected too, as it is not always clear
  • The logistic regression analysis does not make sense, there needs to be greater clarity around what you are stating and what factors you have explored (did you only look at body image and school type?, which factor played a major part? - from my initial reading you are suggesting that body image had little impact, but that school type played a major part – it would be good to see a table with each of the factors you included for both sexes (assuming you examined these separately?) and the odds ratio (spss logistic regression odds ratio = exp(b)) of each factor from within the logistic regression analysis. Also did you include BD, RSS or even age of student in the regression or other factors? If you included sex in the logistic regression, rather than do these separately, you may find some interesting things too
  • Suggest remove a section from your current statement as outlined below as it is redundant or becomes part of the story when examining the factors… (But need to also include the Logistic Regression table with all factors – (there are some great examples which have been provided from other texts on how this might look). This will also help to demonstrate what other factors have contributed or need to be examined further in the future…. i.e. time spent on social media, parent’s income, etc. which you have highlighted in the discussion section.

In the logistic regression analysis, the model containing body image and school type between both sexes, was found to be significant for male adolescents [X2(2)=86.101; p=0.000; R2 =0.222] and female adolescents [X2(2)=188.539; p=0.000; R2 =0.416].

It must be noted that the 22.2% and the 41.6% Variance is only indicating that school type makes up the certain percentage of the whole finding. For example, only 22.2% of the factors that impact male students is from school type, which means 78.8% is made up from some other factors – this seems to be interpreted differently by the authors in the results and discussion section… and it only is telling some of the whole story… hence the need for more information and the table to decipher what it all means…

Discussion

  • This will need revising after re-doing or providing greater context to the logistic regression analysis

Abstract

  • This will need to be revised accordingly when body of the text is revised.

Overall, great study and very important!

Please don’t be disheartened, it is coming along, it just needs some extra work…

Author Response

We thank the reviewers for the suggested changes, which we believe have improved the quality and impact of the manuscript. We have made revisions according to their comments and suggestions, as described below

Introduction:

1. Replace ‘relation’ with ‘relationship’

R: The change has been made.

     Methods:

  1. Should be third person rather than first person language – this is observed elsewhere and should be avoided. Please use third person.

R: The manuscript has been corrected to use the third person.

  1. Methods introduction should also describe the type of study i.e. longitudinal, cross sectional etc. Please provide more detail – I can see some of this in under ‘participants’, please move this under the methods introduction\

R: The sentence has been corrected (2. Method)

  1. ‘Both the adolescents and their parents or guardians were given the informed consent’ is better described as ‘Both the adolescents and their parents or guardians were asked to provide written informed consent’

R: The sentence has been corrected (ethical approval)

  1. ‘trained to clear up all their doubts’ should read ‘trained to support students and address any questions or doubts which the students may have had’

R: The sentence has been corrected (2.4 procedures)

Results

  1. ‘Table 1 shows the RSS and BSQ scores, by sex and school type. In adolescents from private schools, we observed that the mean score in the RSS was similar between the sexes, with no statistically significant differences.’ Should read ‘In adolescents from private schools, we observed that the mean score in the RSS was similar between the sexes, with no statistically significant differences, as outlined in Table 1’. Best to not start a paragraph with Table xxx… This is also observed for Figure 1, Figure 3 – please change this

R: The change has been made in all the tables and figures.

  1. Figure 2 – the x-axis labels are cumbersome – change to High, Medium and Low only with 1 overarching label ‘Self-esteem’, much like the y-axis label ‘percentages’ (Do the same for all figures)

R: The change has been made in all the figures.

  1. Change ‘We further observed that the chance of a female adolescent with BD presenting low self-esteem was 3.85 times higher (CI 95% 2.12-6.99) and 2.83 times higher for males (CI 95% 1.22-6.58)’ to be ‘It was observed female adolescents with BD had 3.85 (CI 95% 2.12-6.99) times higher odds of presenting with low self-esteem than females without BD, which was observed to be 2.83 (CI 95% 1.22-6.58) times higher among males with BD. – This is an issue as it is not clear, but what I think you are saying is if we have two females, one with BD one without, then the Female with BD is 3.85 times more likely to and Esteem issues that the other female without BD?

R: The sentence has been corrected. It refers to the odds of an adolescent with BD having low self-esteem as compared with an adolescent without BD. Adolescents with BD present higher odds of developing low self-esteem than those without BD

  1. Please ensure the above statement is correct in the text of the paper and ensure all other statements made like the above are corrected too, as it is not always clear

R: The correction has been made for students of public and private schools.

  1. The logistic regression analysis does not make sense, there needs to be greater clarity around what you are stating and what factors you have explored (did you only look at body image and school type?, which factor played a major part? - from my initial reading you are suggesting that body image had little impact, but that school type played a major part – it would be good to see a table with each of the factors you included for both sexes (assuming you examined these separately?) and the odds ratio (spss logistic regression odds ratio = exp(b)) of each factor from within the logistic regression analysis. Also did you include BD, RSS or even age of student in the regression or other factors? If you included sex in the logistic regression, rather than do these separately, you may find some interesting things too

R: A new binary logistic regression model was made with the variables: school, sex, level of body satisfaction and age range. We observed that the model was significant and the variables 'adolescents who were students from public schools’ and ‘adolescents with severe BD’ were significant predictor variables of low self-esteem in adolescents. A table was also included with information on all the variables (Table 3).

  1. Suggest remove a section from your current statement as outlined below as it is redundant or becomes part of the story when examining the factors… (But need to also include the Logistic Regression table with all factors – (there are some great examples which have been provided from other texts on how this might look). This will also help to demonstrate what other factors have contributed or need to be examined further in the future…. i.e. time spent on social media, parent’s income, etc. which you have highlighted in the discussion section.

In the logistic regression analysis, the model containing body image and school type between both sexes, was found to be significant for male adolescents [X2(2)=86.101; p=0.000; R2 =0.222] and female adolescents [X2(2)=188.539; p=0.000; R2 =0.416].

It must be noted that the 22.2% and the 41.6% Variance is only indicating that school type makes up the certain percentage of the whole finding. For example, only 22.2% of the factors that impact male students is from school type, which means 78.8% is made up from some other factors – this seems to be interpreted differently by the authors in the results and discussion section… and it only is telling some of the whole story… hence the need for more information and the table to decipher what it all means…

R: The section was removed and replaced with the data of the regression carried out.

Discussion

  1. This will need revising after re-doing or providing greater context to the logistic regression analysis

R: The Discussion was corrected in line with the corrections to the binary logistic regression

Abstract

  1. This will need to be revised accordingly when body of the text is revised.

 R: The Abstract was corrected

Reviewer 2 Report

Dear authors,

The manuscript is easy to read and to understand but I have some major recommendations that should be addressed.

The introduction is very poor and short. It does not cover the huge literature about the topic and all the knowledge that is already known.

The ethical information in the methods section should be moved to a subsection called "ethical aspects".

A description of the sample must be included at the beginning of the results.

p-value cannot be 0.000 it should say >0.001

In Figure 1 is difficult to read the symbols that are above the columns. It must be improved.

Almost all the references are very old. You should do a search for more recent studies about the topic.

Limitations of the study and future research information must be included in the discussion.

Do not use acronyms in the conclusion.

Author Response

We thank the reviewers for the suggested changes, which we believe have improved the quality and impact of the manuscript. We have made revisions according to their comments and suggestions, as described below:

Revisor 2:

  1. The introduction is very poor and short. It does not cover the huge literature about the topic and all the knowledge that is already known.

R: The Introduction has been modified.

  1. The ethical information in the methods section should be moved to a subsection called "ethical aspects".

R: The ethical aspects have been moved to the end of the article.

  1. A description of the sample must be included at the beginning of the results.

R: A description of the sample has been included.

  1. p-value cannot be 0.000 it should say >0.001

R: Modified throughout the manuscript

  1. In Figure 1 is difficult to read the symbols that are above the columns. It must be improved.

R: The figure was corrected.

  1. Almost all the references are very old. You should do a search for more recent studies about the topic.

R: More recent references were included.

  1. Limitations of the study and future research information must be included in the discussion.

R: This datum was included at the end of the Discussion

  1. Do not use acronyms in the conclusion.

R: Modified as requested.

Round 2

Reviewer 1 Report

Well, done. No further suggestions required.

Great work!

Reviewer 2 Report

Dear authors,

Thank you for addressing my comments.

Kind regards